# The Relationship Between Nursing Work Environment and Breastfeeding Prevalence During Child Hospitalization: A Multicenter Study

**DOI:** 10.3390/healthcare12242574

**Published:** 2024-12-21

**Authors:** Martina Batino, Jacopo Fiorini, Francesco Zaghini, Valentina Biagioli, Simona Frigerio, Alessandro Sili

**Affiliations:** 1Department of Biomedicine and Prevention, University of Rome “Tor Vergata”, Via Montpellier, 1, 00133 Rome, Italy; 2Department of Nursing Professions, University Hospital of Tor Vergata, Viale Oxford, 81, 00133 Rome, Italy; 3Department of Medical and Surgical Sciences—DIMEC, University of Bologna, Via Massarenti 9, 40138 Bologna, Italy; 4Nursing Department, University Hospital City of Science and Health, Corso Bramante, 88, 10126 Turin, Italy

**Keywords:** breastfeeding, nursing work environment, nurse workload, pediatric hospitalization, quality of working life

## Abstract

**Background/Objectives**: Breastfeeding during pediatric hospitalization is often challenging, especially in a setting where nursing work environments can affect breastfeeding support. This study examines the relationship between nursing work environments and the prevalence of breastfeeding during child hospitalization, focusing on aspects such as nursing workload, stress levels, and quality of work life (QoWL). **Methods**: A cross-sectional multicenter study was conducted in Italian pediatric hospitals from October 2023 to January 2024. Each ward head nurse completed a form daily for 30 consecutive days to record the number of children breastfed. Nurses’ workload, stress, and QoWL were measured using validated self-report questionnaires. Multivariate regression was employed to examine the associations between organizational variables and breastfeeding prevalence. **Results**: A total of 256 nurses from low- and medium-intensity pediatric wards completed the survey (86.7% female, mean age = 39.2, SD = 9.96). Nurses reported low stress levels (M = 2.74; SD = 0.54) and a good QoWL (M = 3.03; SD = 0.50) despite high workloads (M = 3.34; SD = 0.95). The breastfeeding prevalence mean was 1.12 (SD = 1.38), which was highest in surgical units. Regression analysis revealed that higher breastfeeding prevalence was associated with nurses’ female gender (β = 0.17), clinical setting, and higher levels of nurses’ QoWL (β = 0.14) (R^2^ = 0.23; *p* < 0.001). **Conclusions**: To enhance breastfeeding support during child hospitalizations, healthcare policies should address workload management, stress reduction, and work–life balance. Future longitudinal research should expand to other clinical settings and include detailed patient data to understand these relationships.

## 1. Introduction

According to the World Health Organization (WHO), breastfeeding is crucial to ensuring child and maternal health. Exclusive breastfeeding is recommended for the first six months after birth and should continue for at least one year [1]. Despite its numerous benefits for mother and their children [2,3], breastfeeding rates decline sharply soon after birth [4]. Globally, the breastfeeding rate stands at only 48% [5]. The United Nations International Children’s Emergency Fund (UNICEF) and the World Health Organization (WHO) have reported through the Global Breastfeeding Scorecard that only 40% of children worldwide are exclusively breastfed, and in only 23 countries does the breastfeeding rate exceed 60% [6]. In Italy, the latest data reveal that only 46.7% of children are exclusively breastfed, with significant regional variability; breastfeeding rates are notably lower in southern regions compared to central-northern regions [7]. Investing in breastfeeding support should be a priority to not only prevent infant deaths [8] but also a range of health issues, including respiratory diseases [9], gastrointestinal infection [10,11], childhood obesity risk [12,13], and acute lymphoblastic leukemia [14]. Breastfeeding also benefits mothers by reducing the risk of ovarian and breast cancer [15,16]. Moreover, breastfeeding can lower costs for healthcare organizations and parents, while indirectly contributing to poverty reduction, economic growth, and decreased social inequalities [17,18,19].

Breastfeeding is a complex phenomenon influenced by multiple factors, including structural aspects [20]—such as women’s increasing integration into the work context or cultural influences—as well as individual aspects, such as previous experiences of breastfeeding difficulties, are involved [21,22]. Therefore, the process of initiating, establishing, and maintaining breastfeeding can sometimes become anxiety-provoking and stressful, especially in situations such as a child’s hospitalization [23].

Hospitalization can disrupt breastfeeding due to various clinical conditions in children, such as respiratory diseases [24,25], urinary tract infections [26,27,28,29], or gastrointestinal infections [30,31,32]. Additionally, the working conditions of nursing staff may influence breastfeeding practice for hospitalized children. In 2014, Hallowell et al. found that a staffing ratio of 1 standard deviation higher, adjusted for acuity, was linked to a 2% increase in the number of infants receiving breastfeeding support [33]. This highlights the indirect effects of nursing environments on maternity and child health. Additionally, nurses’ ability to educate and support mothers about breastfeeding during hospitalization can directly influence the continuity of breastfeeding practices [34]. This organizational theme is newly studied for nurses [35], especially after the recognition of breastfeeding as one of the pediatric nursing-sensitive outcomes in low- to medium-complexity care settings [36,37]. Although researchers have examined metrics such as mortality rates or infection control in relation to organizational variables, the nuanced impact of nursing staff dynamics on breastfeeding as a behavioral and health-promoting activity remains underexplored. This novel angle could enhance our understanding of how nursing environments shape family-centered care in pediatric settings [38,39]. The emphasis on the prevalence of breastfeeding among hospitalized children is both new and significant. Traditionally, research on breastfeeding has focused on healthy, full-term infants in the Neonatal Intensive Care Unit (NICU) or in community settings [40]. By examining hospitalized children in low and medium care, we can highlight an underexplored population where breastfeeding could be disrupted by medical interventions and organizational barriers. Addressing breastfeeding support in this context, along with related organizational factors such as nursing workload, stress, and quality of work life, marks an evolution in pediatric nursing care, moving toward holistic, family-centered practices. It will help recognize that breastfeeding is critical for health outcomes, even during hospitalization [41]. This perspective calls for health organizations to enhance the infrastructure for breastfeeding support and to improve training for nursing staff, ensuring that hospitalized children and their mothers can sustain breastfeeding practices despite clinical challenges. Unfortunately, children and mothers often need to adapt to the timing of care provision to align with hospital practices. This means that healthcare organizations must provide support and education about breastfeeding to mothers when their children are admitted to the hospital [42]. Therefore, it is essential to examine how the nursing work environment is associated not only with nurses’ stress and organizational well-being but also with the prevalence of breastfeeding of hospitalized children, as well as to identify organizational strategies for improvement.

## 2. Background

Breastfeeding is crucial to ensuring child and maternal health. However, within healthcare organizations, mothers encounter various barriers to maintaining breastfeeding, such as identifying the right time and finding a dedicated space in the ward for breastfeeding, as well as concerns about potentially causing clinical harm to their child [43]. Previous research highlights that the prevalence of breastfeeding during a child’s hospitalization can be influenced by multiple factors, including nursing staffing levels [44] and nurses’ perceived workload [45]. The nurse-to-patient ratio is commonly used as a nurse staffing index, reflecting the number of nurses on shift and caring for patients. Studies have shown that this ratio affects nurses’ work engagement and burnout [46,47]. Insufficient staffing often leads to excessive workloads, errors, and a reduced capacity to provide attentive and personalized care [48,49]. This situation can also contribute to a perceived decline in the quality of work life (QoWL) among staff [50,51] and a lower capability to provide breastfeeding support.

While the influence of nursing workloads on care quality and clinical outcomes has been studied, there is a lack of research directly linking the nursing work environment to the prevalence of breastfeeding during pediatric hospitalizations [41]. Nursing workloads are a multidimensional concept that depends on various factors, including the amount of time available for caring, staffing levels and skill mix, and the emotional and physical demands and complexity of care required [52,53]. Nursing workloads are “the amount of work a nurse performs within a specific period, the number of tasks required, the work carried out on patients and administrative tasks” [54,55]. Two previous studies identified the relationship between nursing workloads and breastfeeding support [56,57]. High workloads and a poor nurse-to-patient ratio may influence job satisfaction [58], QoWL [59], and stress levels [60], hindering nurses from effectively promoting and supporting breastfeeding [56,57]. Furthermore, the organizational context plays a critical role in shaping nurses’ perceptions and emotions [61], which can subsequently impact care outcomes [62]. However, the specific effects of the nursing work environment on breastfeeding support during pediatric hospitalization remain largely unexplored. This gap is particularly significant given the recent recognition of breastfeeding as a nursing-sensitive outcome [37]. Evaluating this relationship is crucial for developing strategies that foster a healthy work environment for nurses, ultimately supporting mothers in maintaining breastfeeding and improving pediatric care outcomes.

Nurses are a unique group within the healthcare workforce as they face multiple emotional demands [63,64,65] that affect their QoWL [66] and work-family conflict [67]. Several studies have investigated how organizational context variables impact nurses’ QoWL and care outcomes in healthcare settings [68]. However, none has specifically addressed factors such as workloads, stress, and QoWL regarding breastfeeding support during child hospitalization. Research indicates that organizational well-being significantly affects professionals’ performance [69], influencing the perceived quality of care services [70]. In a healthcare setting, a head nurse’s leadership style that promotes a positive organizational climate leads to greater motivation and engagement among nurses. As a result, nurses perform better in their clinical activities, and patients perceive improved quality of care provided [68,71,72]. The organizational context has already been identified as a predictor of nurses’ performance, influenced by various factors such as the nursing staff, skill mix, and workloads [73,74]. These factors can consequently affect nurses’ QoWL [60,75] and nursing-sensitive outcomes [76], including breastfeeding [77].

Therefore, this study aims to examine the relationship between the nursing work environment—shaped by factors such as nursing workload, stress, and nursing quality of working life—and the prevalence of breastfeeding during a child’s hospitalization. In particular, we hypothesized that in clinical settings where nurses experience lower workloads, reduced stress, and greater QoWL, the number of patients who are breastfed is higher.

## 3. Materials and Methods

### 3.1. Study Design

Between October 2023 and January 2024, a cross-sectional multicenter study was conducted across three Italian pediatric hospitals. This study adhered to the Strengthening the Reporting of Observational Studies in Epidemiology (STROBE) Checklist for its methodology.

### 3.2. Study Settings

The study was conducted in pediatric non-intensive medical and surgical wards of three hospitals. Each hospital has a minimum of three pediatric departments and ten wards, with at least twelve beds in each ward. Nurses working in these wards followed a shift schedule divided into three time slots: morning (7:00 a.m.–2:00 p.m.), afternoon (2:00 p.m.–9:00 p.m.), and night (9:00 p.m.–7:00 a.m.).

### 3.3. Sampling and Participants

A convenience sample of nurses was enrolled for the study if they had worked for at least six months in pediatric wards and provided direct care to children (patients aged 0–16 years), without restrictions based on shift type or contract status (full-time/part-time or fixed-term/open-ended). However, novice nurses with less than six months of experience, as well as nursing managers, head nurses, and nursing assistants, were excluded.

According to a national survey of the nursing working environments, the mean quality of working life was reported as 2.82 (standard deviation of 0.58) [78]. Assuming a higher quality of working life in the pediatric setting, we anticipated a mean of 2.95, a significance level (alpha) of 0.05, and a statistical power of 80%. Thus, the estimated sample size required was 156 participants. To account for an anticipated participant loss of 20% (e.g., due to incomplete surveys), at least 187 participants should be enrolled to ensure the study maintains its statistical power.

### 3.4. Data Collection

Data were collected through a self-report questionnaire enclosed using a Google Form survey, shared via a link sent to the personal nurses’ emails. A total of 351 emails were sent, and weekly reminders were sent to encourage participation. The first page of the survey described the study’s aim and requested informed consent from the participants. The questionnaire was fully anonymous. Additionally, a specific ward form was used to collect daily staff levels and breastfeeding prevalence over a thirty-day observation period.

### 3.5. Instruments

Validated scales, as described in Table 1, and sociodemographic and work characteristics (e.g., department affiliation, years of service in that department) comprised the self-report questionnaire for nurses. Each instrument’s final score is calculated as the mean (SD), and the higher the score, the greater the perceived level of the construct.

The ward form was specially developed to collect breastfeeding information and nursing staffing. Each ward head nurse filled in the form daily for 30 consecutive days, recording the number of nurse staff scheduled for each of the three shifts and the prevalence of breastfeeding in the ward (number of children breastfed in the ward with exclusive or mixed breastfeeding). Before data collection, head nurses were trained to complete the form accurately.

### 3.6. Ethical Statements

The study followed the Declaration of Helsinki principles [83] and was approved by the Ethics Committee of the University Hospital of Rome “Tor Vergata” (Prot. No. RS143.21). All participants were assigned an alphanumeric code to ensure anonymity, and data were analyzed in aggregate form. Online data storage was created and protected by a double identification system (personal ID and secure code), which was accessible only to the researchers.

### 3.7. Data Analysis

Frequencies and percentages were used to summarize the qualitative characteristics of the data, while central tendency indices (means and medians) and dispersion (standard deviation, minimum, and maximum) were calculated to describe quantitative data. The distribution of continuous data was examined for normality using the Shapiro–Wilk test. The one-way ANOVA test was used to compare the differences between the clinical settings and the quantitative variables studied. Post hoc analysis for significant ANOVA results was performed using Tukey’s Honest Significant Difference (HSD) test. Based on the univariate and Pearson correlation analyses, a stepwise multivariate regression model was conducted, using the prevalence of breastfeeding as the outcome and working context and demographic variables as independent variables. Multicollinearity was evaluated using the variance inflation factor (VIF). The internal consistency of the scales was calculated using Cronbach’s Alpha test. All analysis was performed with the statistical package SPSS Ver 25^®^. The significance level was set at *p* < 0.05.

## 4. Results

### 4.1. Sociodemographic and Work Characteristics of the Sample

Out of 351 nurses invited to participate in the study, 265 completed the questionnaire, resulting in a response rate of 75.5%. Overall, 86.7% (n = 222) of participants were female, with an average age of 39.2 years (SD = 9.96). More than half of the sample (55.1%, n = 141) were married (Table 2). Additionally, 65.2% (n = 165) of nurses held a degree in nursing, and they had an average of 14.3 years (SD = 9.75) of work experience.

### 4.2. Descriptive Analysis

Overall, each ward had 15.49 (SD = 8.06) beds and 13.19 (SD = 7.09) inpatients, with an average length of hospitalization of 12.10 (SD = 9.57) days. The nurse-to-patient ratio in these units varied from 1:3 during day shifts to 1:4 during night shifts. Nurses reported low stress levels and a good quality of working life, despite slightly higher workloads. Additionally, at least one breastfed child was noted in the observations recorded over the 30 days, with a breastfeeding prevalence ranging from 0 to 4 patients depending on the clinical setting. The one-way ANOVA results showed statistically significant differences in breastfeeding prevalence according to nursing clinical settings (F = 20.39; *p* < 0.001; Table 3). In particular, the highest prevalence of breastfed children was recorded in the surgical setting (M = 1.93; SD = 1.95) and the lowest in the hematology setting (M = 0.38; SD = 0.30), as shown by the Tukey post hoc test (indicated using superscript letters in the Table).

### 4.3. Associations Among Study Variables

Pearson’s correlation analysis revealed that as nurses’ perceived stress increased, their Quality of Work Life (QoWL) declined (Table 4). Moreover, the prevalence of breastfeeding in the ward was significantly associated with QWI, QoWL, and nurses’ gender. Specifically, a higher prevalence of breastfeeding was observed in those wards with predominantly female nursing staff and where nurses reported higher QoWL and workload.

The linear regression analysis revealed that 23% of the variability in breastfeeding prevalence during the child hospitalization could be attributed to clinical settings, QoWL, and nurses’ female gender (R^2^ = 0.23; *p* < 0.001). In particular, the prevalence of breastfeeding was independently associated with nurses’ female gender, type of clinical setting, and higher levels of QoWL (Table 5).

## 5. Discussion

This study investigated the relationship between the nursing work environment, influenced by nurses’ gender, stress, and quality of working life levels, and the prevalence of breastfeeding during a child’s hospitalization. The findings highlight a relationship between a supportive nursing work environment and the prevalence of breastfeeding practices. This supports the hypothesis that healthcare organizations fostering nursing working environments enable nurses to perceive higher, which translates into a better quality of care, including breastfeeding prevalence.

A notable finding is the breastfeeding prevalence variation across clinical settings, with the highest rates observed in the surgical area. Multiple factors can explain this disparity, such as the age of children and clinical conditions [84]. While previous research emphasizes the negative influence of nurses’ workload on patient outcomes [75], our research suggests that in pediatric settings where nurses report higher QoWL, the breastfeeding prevalence increases even if they experience high workloads and stress levels. Nurses’ QoWL can be influenced by various factors [85], such as job stress and burnout [66]. These include effective coping strategies and certain moderating aspects of work conditions, such as a favorable nurse-patient ratio [33] or employment in a public hospital [76]. These factors may help explain our results, demonstrating that, even in demanding environments, nurses can maintain a good QoWL and prioritize patient care, especially for pediatric patients.

Recognizing breastfeeding as a nursing-sensitive outcome [37], we hypothesized that workloads could impact breastfeeding prevalence. However, our results underline a positive relationship between these variables at univariate analysis and a non-significant association at the regression analysis. Previous research has identified the negative influence of workloads on patient outcomes [73,74]. As a result, several organizational interventions have been implemented to support nurses, guarantee adequate nurse workloads during shifts, and improve the quality of patient care. These include redesigning work systems to adapt physical working environments, ensuring adequate staffing levels, and encouraging cooperation and collaboration among staff. Additionally, encouraging nurses to actively participate in identifying appropriate solutions for organizing their work helps to guarantee manageable workloads during shifts and promote the quality of care provided to patients [86].

It is worth noting that our results at univariate analysis may suggest that breastfeeding practices during child hospitalization may increase the nurses’ workloads. In particular, surgery was the clinical setting where the highest prevalence of breastfeeding was observed and where nurses reported the greatest workload. If mothers breastfeed, nurses may have an active role in preparing and managing feeding logistics, educating, and supporting them physically and psychologically, thus reducing their time for other clinical activities, such as discharge planning [23]. Future studies should better evaluate the relationship between nurses’ workload and breastfeeding prevalence in clinical settings, trying to understand if breastfeeding can increase the nurses’ workloads, using a longitudinal design.

Moreover, workloads were significantly associated with stress levels, which were not significantly correlated to breastfeeding in our sample. However, previous research has shown that high stress levels can reduce nurses’ performance in delivering care, leading them to prioritize only certain activities, such as intravenous administration [87]. As a result, important aspects of care, such as breastfeeding support for mothers and their children, may be neglected or only partially completed by nurses, resulting in missed nursing care [87,88]. In such situations, implementing stress-reduction interventions could potentially enhance the quality of nursing care, which may indirectly improve breastfeeding outcomes [89].

In particular, in this study, nurses reported a generally positive QoWL and lower stress levels, despite high workloads. This situation suggests several possibilities. During their clinical practice, nurses may employ effective coping mechanisms, which could enhance their resilience in the workplace [35,90]. Also, their personal and professional values may positively influence their working condition [91,92]. Maintaining a good ratio of nurses to patients during shifts and having a supportive working environment may help moderate the nurses’ perceptions of their QoWL even when there are high workloads [85]. This can contribute to high-quality patient care and improve breastfeeding prevalence, supporting findings from similar research [86,93]. In addition, factors such as being female and a mother may play a role in motivating nurses to support breastfeeding, as they easily recognize the protective effects of breastfeeding on a child’s health and the mother’s breast cancer incidence [87,94].

Lastly, the positive correlation between QoWL and breastfeeding prevalence emphasizes the importance of work–life balance for nurses to enhance nursing-sensitive outcomes. A higher QoWL is associated with increased engagement and a greater capacity to provide emotional and practical support to mothers [88,95]. Furthermore, research in healthcare organizations consistently highlights the need for policies that promote nurses’ well-being, including manageable workloads, flexible scheduling, and access to mental health resources [96]. Such measures are essential components of global strategies to promote breastfeeding. The implications of our findings align with these global strategies, which aim to enhance breastfeeding rates by improving healthcare policies. For example, creating breastfeeding-friendly environments [97], ensuring adequate nurse–patient ratios, and implementing comprehensive lactation training programs for nurses could significantly increase breastfeeding prevalence worldwide [33]. These policies not only improve the quality of work life for nurses but also enhance the quality of nursing care. This, in turn, could support breastfeeding during child hospitalizations [98].

### Limitations

The study has several limitations. First, this was a cross-section study; therefore, data cannot be used to infer causality since a temporal sequence cannot be established. Second, we used a convenience sampling method, introducing potential selection bias. Third, data were collected in low- and medium-intensity care units, focusing on children aged 0 to 16 who were admitted, without specific information on the patients’ age during the study period. Additionally, we collected information on breastfeeding prevalence rather than maintenance. Thus, some children may have never been breastfed from birth or could have stopped breastfeeding due to maternal choice. Although data were collected during the winter season, the period with the highest admission rates in pediatric wards, the limited observation time of only 30 days may have affected the findings. Finally, the low R² value (23%) in the regression analysis may indicate that other important factors influencing breastfeeding prevalence, such as the child’s health conditions or maternal education level, were not captured in the study. Future research should expand to other clinical settings and examine detailed characteristics of mothers and patients, such as age and health condition. Additionally, a longitudinal design should be employed to establish causality and use a longitudinal design to establish causality.

## 6. Conclusions

Overall, this study highlights the significant association between work conditions and breastfeeding prevalence and emphasizes the importance of supporting nursing staff. Fostering a supportive organizational environment is essential not only for improving nurses’ quality of work life but also for increasing breastfeeding rates. To achieve this, healthcare organizations must implement policies that effectively manage workloads, reduce stress, and promote work–life balance. These policies would enable nurses to improve nursing-sensitive outcomes, such as providing high-quality breastfeeding support during child hospitalizations, ultimately benefiting maternal and child health. Further studies are needed to investigate additional contextual variables and expand these findings to different pediatric settings.

## Figures and Tables

**Table 1 healthcare-12-02574-t001:** Investigated variables and related instruments used.

Variable	Scale	Authors	N. Item	Dimension	Response	Validation Study Cronbach’s α	Study Cronbach’s α
Workload	QWI (Quantitative Workload Inventory)	Spector et al., 1998 [79]; Barbaranelli et al., 2013 [80]	4	Single	5-point Likert scale (from 1 = never or not at all to 5 = very often/always)	0.74	0.86
Work-related Stress	Health and Safety Executive Indicator tool HSE—IT (STRESS)	Marcatto et al. (2011) [81]	19	3 (demand, control, support)	4-point Likert scale (from 1 = never to 4 often)	0.85; 0.80; 0.92	0.86
Nursing Quality Working Life	Nursing Quality of Life Scale (NQoLs)	Sili et al. (2018, 2022) [78,82]	14	2 (working QoL, social QoL)	4-point Likert scale (from 1 = not at all satisfied to 4 very satisfied)	0.89; 0.81	0.89

**Table 2 healthcare-12-02574-t002:** Sample sociodemographic and work variables (n = 265).

Variables	N	%	M	SD
Age			39.2	9.96
Gender				
Female	222	86.7		
Male	34	13.3		
Non-binary	-	-		
Marital status				
Single	98	38.3		
Divorced	16	6.30		
Married	141	55.1		
Widow	1	0.4		
Qualification level				
Regional diploma	48	18.8		
University diploma	41	16		
Bachelor’s degree	167	65		
Working years			14.3	9.75
Daily hours			0.98	3.22
Weekly Overtime hours			7.11	1.98
No. of absences in the last six months			7.58	9.15

**Table 3 healthcare-12-02574-t003:** Descriptive statistics of variables under study.

	Total (N = 256)		Hematology (N = 89)		Surgery (N = 63)		Pediatric (N = 104)		Anova One-Way	
	M (SD)	Range	M (SD)	Range	M (SD)	Range	M (SD)	Range	F	*p*
BREAS	1.12 (1.38)	0–4.37	0.38 (0.30) ^c^	0.07–1.00	1.93 (1.95) ^a^	0–4.37	1.27 (1.23) ^b^	0–3.30	29.39	<0.001
QWI	3.34 (0.95)	1.25–5.00	3.32 (0.96) ^ab^	1.50–5.00	3.58 (1.00) ^a^	1.25–5.00	3.23 (0.90) ^b^	1.50–5.00	2.85	0.06
STRESS	2.74 (0.54)	1.37–4.58	2.71 (0.64)	1.48–4.58	2.81 (0.52)	1.53–4.05	2.74 (0.47)	1.37–3.79	0.64	0.53
QoWL	3.03 (0.50)	1.43–4.00	3.03 (0.50)	1.64–4.00	3.07 (0.43)	2.00–3.93	3.01 (0.55)	1.43–4.00	0.25	0.78

Note: BREAS = Breastfeeding, QWI = Quantitative Workload Inventory, QoWL = Quality of Working Life; M = Mean; SD = standard deviation; means with different superscript letters are significantly different at the Tukey post hoc test.

**Table 4 healthcare-12-02574-t004:** Pearson’s correlations between variables under study.

Variable	Gender (Female)	STRESS	QWI	BREASTF	QoWL	Age
STRESS	−0.052	—				
QWI	−0.003	0.594 **	—			
BREASTF	0.202 **	0.120	0.150 *	—		
QoWL	−0.043	−0.223 **	−0.043	0.125 *	—	
Age	−0.106	0.025	−0.145 *	−0.060	−0.098	—
Working years	−0.151 *	0.022	−0.121 *	−0.030	−0.085	0.887 **

Note: Female gender; BREAS = Breastfeeding, QWI = Quantitative Workload Inventory, QoWL = Quality of Working Life; ** *p* < 0.001; * *p* < 0.05.

**Table 5 healthcare-12-02574-t005:** Linear regression analysis predicting the prevalence of breastfeeding.

Model	R	R^2^	Adjusted R^2^	RMSE	Sum of Squares	df	Mean Square	F	*p*
	0.507	0.257	0.233	1.217	126.691	8	15.836	10.695	0.001
Coefficients	Unstandardized		Standard Error		Standardized ᵃ	t	*p*	Collinearity statistics	
								Tolerance	VIF
(Intercept)	−2.190		0.847			−2.586	0.010		
QoWL	0.393		0.158		0.142	2.487	0.014	0.779	1.284
STRESS	0.265		0.181		0.104	1.465	0.144	0.773	1.294
QWI	0.081		0.103		0.056	0.790	0.430	0.779	1.284
Gender (Female)	0.677		0.230		0.166	2.941	0.004	0.974	1.027
Age	−0.010		0.008		−0.071	−1.233	0.219	0.956	1.046
Setting-Surgery	1.420		0.205			6.921	<0.001	0.977	1.024
Setting-Pediatric	0.950		0.179			5.306	<0.001	-	-

Note: ᵃ Standardized coefficients can only be computed for continuous predictors; df = degree of freedom; VIF = Variance inflation factor; QoWL = Quality of Working Life; QWI = Quantitative Workload Inventory.

## Data Availability

The data presented in this study are available upon request from the corresponding author due to privacy or ethical restrictions.

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
