# Peer review of "The Relationship Between Nursing Work Environment and Breastfeeding Prevalence During Child Hospitalization: A Multicenter Study"

_healthcare, 2024, doi:10.3390/healthcare12242574_

Round 1
Reviewer 1 Report
Comments and Suggestions for Authors
Comments have been attached as a separate file

Author Response
Comments 1: General Impression
The study addresses an important topic: “the relationship between nursing work environments and breastfeeding prevalence during pediatric hospitalizations.” Although the topic is significant and well-aligned with public health goals, methodological rigour, clarity, and discussion need attention.
Response 1: Thank you for your valuable comments. We hope that the revisions have improved the manuscript.
Comments 2: Abstract
The abstract effectively summarizes the study's objectives, methodology, and key findings, making it easy for readers to understand the focus of the research. It highlights the relevance of nursing work environments in influencing breastfeeding prevalence, an important aspect of healthcare practice and policy. However, the abstract does not provide precise numerical data to support the findings. Including more specific data points, like the average breastfeeding prevalence and significant associations (e.g., QoWL and breastfeeding prevalence, R² = 0.23, p < 0.001), would help substantiate the claims and strengthen the results.
The description of the methodology for the abstract is too general. For example, it mentions a "cross-sectional multi-centre study" but does not specify the sample size (e.g., 256 nurses) or how breastfeeding prevalence was measured. Provide a concise overview of the study design, sample, and key measures used (e.g., validated scales for workload, stress, and QoWL).
Response 2: Thank you for your valuable comments. You can find more details on methods and results to support the study design and the findings as suggested.
Comments 3: Introduction
The introduction emphasizes the importance of breastfeeding for maternal and child health while addressing challenges such as declining rates and organizational barriers. It provides context through global trends and regional variability in Italy, helping readers understand the broader implications of the research.
However, while the introduction highlights the importance of breastfeeding and organizational factors, it does not identify the specific gaps this study addresses. The authors must explicitly state the research gap as part of the introduction.
For instance:
- How does this study build on or differ from prior research linking nursing environments to patient outcomes?
- Why is this focus on breastfeeding prevalence in hospitalized children novel or significant?
The aims of the study are mentioned, but they are vague and scattered across paragraphs. Authors may consider outlining clearly, the study objectives in a focused statement at the end of the introduction.
Response 3: Thank you for your valuable comments. The literature gap is “Although researchers have examined metrics such as mortality rates or infection con-trol in relation to organizational variables, the nuanced impact of nursing staff dy-namics on breastfeeding as a behavioral and health-promoting activity remains un-derexplored. This novel angle could enhance our understanding of how nursing envi-ronments shape family-centered care in pediatric settings [38,39]. The emphasis on the prevalence of breastfeeding among hospitalized children is both new and significant. Traditionally, research on breastfeeding has focused on healthy, full-term infants in the Neonatal Intensive Care Unit (NICU) or in community settings [40]. By examining hospitalized children in low and medium care, we can highlight an underexplored population where breastfeeding could be disrupted by medical interventions and organizational barriers”.
To clarify the aim, we have now added a hypothesis at the end of the background: “In particular, we hypothesized that in clinical settings where nurses experience lower workloads, reduced stress and greater QoWL, the number of patients who are breastfed is higher”
Comments 4: Background
The background highlights breastfeeding benefits, challenges during hospitalization, and the impact of nursing factors setting the stage for the study. It also connects nursing-sensitive outcomes and organizational factors, supporting the research focus.
However, the background lacks a clear flow, jumping between breastfeeding challenges and nursing factors. Reorganizing the section first to address breastfeeding challenges, then nursing roles, and finally the impact of workload, stress, and QoWL would improve clarity.
The discussion on workload and stress is superficial. The authors should expand on how these factors hinder breastfeeding support, citing studies linking higher workloads to less time for individualized care.
The background does not identify gaps in the current literature or the study's necessity. It should highlight the lack of research linking nursing work environments to breastfeeding prevalence during pediatric hospitalizations, emphasizing the study's novelty.
Response 4: Thank you for your valuable comments. The section has been revised accordingly. The gap in the literature was also highlighted. “While the influence of nursing workloads on care quality and clinical outcomes has been studied, there is a lack of research directly linking the nursing work environment to the prevalence of breastfeeding during pediatric hospitalizations”… “However, the specific effects of the nursing work environment on breastfeeding support during pediatric hospitalization remain largely unexplored. This gap is particularly significant given the recent recognition of breastfeeding as a nursing-sensitive outcome. Evaluating this relationship is crucial for developing strategies that foster a healthy work environment for nurses, ultimately supporting mothers in maintaining breastfeeding and improving pediatric care outcomes”.
Comments 5: Materials and Methods
The study utilizes established, validated instruments to assess workload, stress, and quality of work life (QoWL), ensuring the reliability of its findings. Conducting the study across three pediatric hospitals strengthens the generalizability of the findings within the Italian healthcare context.
However, the study uses a convenience sampling method, introducing potential selection bias. It is unclear how representative the sample is of the larger nursing population. Discuss the implications of this sampling method on generalizability and consider including a statement about its limitations.
The authors demonstrate adherence to ethical principles, including approval by an ethics committee and informed consent procedures but did not specifically mention the name of the institution that provided the Ethical Clearance. Kindly state the name of the review board.
Response 5: Thank you for your valuable comments. We have addressed the convenience sample among the limitations of the study: “Second, we used a convenience sampling method, introducing potential selection bias”. In the methods, we have also added a sample size calculation: “According to a national survey of the nursing working environments, the mean quality of working life was reported as 2.82 (standard deviation of 0.58) [78]. Assuming an anticipated mean of 2.95, a significance level (alpha) of 0.05, and a statistical power of 80%, the estimated sample size required was 156 participants. To account for an anticipated participant loss of 20% (e.g. due to incomplete surveys), at least 187 participants should be enrolled to ensure the study maintains its statistical power.”
Also, we add the name of the Ethical Committee “University Hospital of Rome Tor Vergata”
Comments 6: Results
The section provides a clear summary of sample demographics and work characteristics, contextualizing the findings. It effectively presents breastfeeding prevalence data and employs appropriate statistical methods, including ANOVA and regression analysis.
Response 6: Thank you for your valuable comments
Comments 7: Discussion
The authors effectively relate the results to existing literature, emphasizing the critical role of QoWL in breastfeeding support. The discussion provides actionable recommendations, such as enhancing QoWL and managing workloads, and positions breastfeeding prevalence as a key nursing-sensitive outcome, aligning with public health priorities.
The low R² value (23%) in the regression analysis is not adequately discussed. This suggests that other important factors influencing breastfeeding prevalence were not captured in the study. Discuss potential unmeasured variables and their relevance to future research.
While the discussion calls for workload reduction and QoWL improvement, it lacks concrete suggestions or examples of successful interventions in similar contexts. Provide specific examples of policies or interventions that could improve breastfeeding support in hospital settings.
The discussion primarily focuses on the Italian context without connecting findings to global breastfeeding challenges or nursing work conditions. Broaden the implications to address how these findings could inform global strategies for improving breastfeeding rates in healthcare settings.
The discussion mentions the cross-sectional design but does not fully explore other limitations, such as potential biases from convenience sampling. Expand the discussion of the limitations to provide a balanced interpretation of the results.
Response 7: Thank you for your comments. We have expanded the limitations as you suggested: “Finally, the low R² value (23%) in the regression analysis may indicate that other important factors influencing breastfeeding prevalence, such as the child’s health conditions or maternal education level, were not captured in the study. Future research should expand to other clinical settings and include detailed characteristics of mothers and patients, such as age and health condition.”
In the discussion, we have now provided some examples of organizational interventions: “As a result, several organizational interventions have been implemented to support nurses, guarantee adequate nurse workloads during shifts, and improve the quality of patient care. These include redesigning work systems to adapt physical working environments, ensuring adequate staffing levels, and encouraging cooperation and collaboration among staff. Additionally, encouraging nurses to actively participate in identifying appropriate solutions for organizing their work helps to guarantee manageable workloads during shifts and promote the quality of care provided to patients”
Moreover, we have now connected findings to global breastfeeding promotion strategies, as suggested: “Furthermore, research in healthcare organizations consistently highlights the need for policies that promote nurses' well-being, including manageable workloads, flexible scheduling, and access to mental health resources [94]. Such measures are essential components of global strategies to promote breastfeeding. The implications of our findings align with these global strategies, which aim to enhance breastfeeding rates by improving healthcare policies. For example, creating breastfeeding-friendly environments [95], ensuring adequate nurse-patient ratios, and implementing comprehensive lactation training programs for nurses could significantly increase breastfeeding prevalence worldwide [33].
Comments 8: Conclusion
The conclusion highlights the link between nursing work environments and breastfeeding prevalence, stressing the importance of QoWL and supportive policies. It offers actionable insights however; it overstates the findings and presents them as definitive. The conclusion should acknowledge the correlational nature of the results and suggest the need for further investigation.
The suggested interventions are too general and lack actionable detail. Offer specific, and evidence-based recommendations.
The conclusion focuses on breastfeeding during pediatric hospitalizations but doesn’t connect the findings to broader healthcare goals. It should expand to discuss how these findings contribute to global breastfeeding promotion strategies.
Response 8: Thank you for your valuable comments. We have now revised the conclusion: ”Overall, this study highlights the significant association between work conditions and breastfeeding prevalence and emphasizes the importance of supporting nursing staff. Fostering a supportive organizational environment is essential not only for improving nurses’ quality of work life but also for increasing breastfeeding rates. To achieve this, healthcare organizations must implement policies that effectively manage workloads, reduce stress, and promote work-life balance. These policies would enable nurses to improve nursing-sensitive outcomes, such as providing high-quality breastfeeding support during child hospitalizations, ultimately benefiting maternal and child health”.
Comments 9: Ethics Committee
Indicate the name of the institution or committee that provided the Ethical Clearance.
Response 9: Thank you for your valuable comments. We insert the name of the Ethical Committee of the University Hospital of Rome, “Tor Vergata”.
Reviewer 2 Report
Comments and Suggestions for Authors
Thank you for giving me the opportunity to evaluate this article. My suggestions regarding the article are attached.
Abstract Section,
- It is written appropriately and reflects the content. It would be appropriate to add the full sample number only to the method.
Introduction Section,
- This section is written in a way that defines each independent variable, but it would be appropriate to summarize the direct and indirect effects of variables related to nurses on breastfeeding in a paragraph to reflect the purpose of the study more clearly.
Method Section,
- In the sample section, how the a priori sample size was calculated, how many nurses they planned to include in the study in total, how many nurses they took in total should be added (a priori sample size calculated according to power and effect size)
- It was nice to make a table regarding the measurement tools used in the study, but the total score of the scales, validity results, what the score obtained from the scales means as the score increases are not specified. This information should be added.
- In the data analysis section, it should be written how the distribution of continuous data is examined. What is used as post-hoc analysis for Anova analysis should be stated. How multicollinearity is evaluated for regression analysis should be added. How autocorrelation is examined for regression analysis should be explained.
Findings Section,
- Generally written appropriately. Post hoc analysis results are not given for the results that differ in Table 3. Post-hoc analysis results should be added. Table 5 (regression table) confidence intervals, DW value, F and P value, R square and corrected R square values ​​should be given.
Discussion Section,
- In general, the findings are discussed with the support of literature. The discussion should be expanded in line with the suggestions in the findings.
References
- Current and appropriate to the subject.
Author Response
Comments 1: General Impression
Thank you for allowing me to evaluate this article. My suggestions regarding the article are attached.
Response 1: Thank you for your time, every suggestion is valuable
Comments 2: Abstract Section
- It is written appropriately and reflects the content. It would be appropriate to add the full sample number only to the method.
Response 2: Thank you for your suggestion. We have now added this information.
Comments 3: Introduction Section
- This section is written in a way that defines each independent variable, but it would be appropriate to summarize the direct and indirect effects of variables related to nurses on breastfeeding in a paragraph to reflect the purpose of the study more clearly.
Response 3: Thank you for your comments. We have now added in the text: “Addressing breastfeeding support in this context, along with related organizational factors such as nursing workload, stress, and quality of work life, marks an evolution in pediatric nursing care, moving towards holistic, family-centered practices. It will help recognize that breastfeeding is critical for health outcomes, even during hospitalization [41]. This perspective calls for health organizations to enhance the infrastructure for breastfeeding support and to improve training for nursing staff, ensuring that hospitalized children and their mothers can sustain breastfeeding practices despite clinical challenges”.
Comments 4: Method Section
- In the sample section, how the a priori sample size was calculated, how many nurses they planned to include in the study in total, and how many nurses they took in total should be added (a priori sample size calculated according to power and effect size)
- It was nice to make a table regarding the measurement tools used in the study, but the total score of the scales, validity results, and what the score obtained from the scales means as the score increases are not specified. This information should be added.
- In the data analysis section, it should be written how the distribution of continuous data is examined. What is used as post-hoc analysis for Anova analysis should be stated. How multicollinearity is evaluated for regression analysis should be added. How autocorrelation is examined for regression analysis should be explained.
Response 4: Thank you for your valuable comments.
We have now added the sample size calculation to the manuscript: “According to a national survey of the nursing working environments, the mean quality of working life was reported as 2.82 (standard deviation of 0.58) [78]. Assuming a higher quality of working life in the pediatric setting, we anticipated a mean of 2.95, a significance level (alpha) of 0.05, and a statistical power of 80%. Thus, the estimated sample size required was 156 participants. To account for an anticipated participant loss of 20% (e.g. due to incomplete surveys), at least 187 participants should be enrolled to ensure the study maintains its statistical power”.
We have revised the table to add more information on the measurement tools. We have also added in the text: “Each instrument's final score is calculated as the mean (SD), and the higher the score, the greater the perceived level of the construct.”
In the data analysis section, we have added: “The distribution of continuous data was assessed for normality using the Shapiro-Wilk test.” We have also specified “Post-hoc analysis for significant ANOVA results was performed using Tukey’s Honest Significant Difference (HSD) test”.
Concerning multicollinearity, we have now added “Multicollinearity was evaluated using the variance inflation factor (VIF)”.
Comments 5: Findings Section
- Generally written appropriately. Post hoc analysis results are not given for the results that differ in Table 3. Post-hoc analysis results should be added. Table 5 (regression table) confidence intervals, DW value, F and P value, R square and corrected R square values ​​should be given.
Response 5: Thank you for your valuable comments, we have now revised the tables accordingly. The Post hoc analysis results have been reported using superscript letters near each means. To better clarify this aspect we have added in the text “as shown by the Tukey post hoc test (indicated using superscript letters in the Table)”.
Comments 6: Discussion Section
- In general, the findings are discussed with the support of literature. The discussion should be expanded in line with the suggestions in the findings.
Response 6: Thank you for your valuable comments. We have now expanded the discussion to explain the findings.
Comments 7: References
- Current and appropriate to the subject.
Response 7: Thank you for your comment, we have carefully read the articles in the literature.
Reviewer 3 Report
Comments and Suggestions for Authors
Very interesting article, can be published after addressing comments:
This article provides important insights into the intersection of nursing environments and breastfeeding outcomes. Refining the discussion and providing actionable recommendations would further enhance its impact. The study makes a solid contribution to the pediatric and nursing literature, and offers a basis for policy development and clinical practice improvements.
Strengths of the article:
Relevance and importance: The study addresses a critical health issue by examining how the nursing work environment influences breastfeeding prevalence, a significant determinant of maternal and child health.
Comprehensive scope: Inclusion of multiple children's hospitals improves the generalizability of the findings.
Validated instruments: The use of established instruments (e.g., QWI, HSE-IT, and NQoLs) strengthens the reliability of the data.
Analytical rigor: The combination of descriptive statistics, Pearson correlations, and multivariate regression analysis is methodologically sound.
Ethical considerations: The study adheres to the Declaration of Helsinki and ensures anonymity and informed consent, while demonstrating ethical rigor.
Suggestions for improvement:
Abstract: The abstract concisely presents the findings, but it could better outline the limitations and implications for practice to provide a holistic summary.
Introduction: While the introduction is informative, adding more recent global and regional data on breastfeeding trends could better contextualize the study.
Literature review: The background section is strong but could benefit from a more critical appraisal of previous studies that have conflicting findings or gaps in addressing the impact of the nursing work environment.
Methods: The sampling strategy relies on convenience sampling, which may introduce bias. Justification for this approach or discussion of its limitations could strengthen the methodology section.
Results:
The tables are clear but could benefit from additional visual aids (e.g., bar graphs or scatter plots) to illustrate key trends and relationships.
Discussion of differences in breastfeeding prevalence across clinical settings could delve into potential cultural or systemic factors influencing outcomes.
Discussion:
The authors highlight the positive association of QoWL with breastfeeding prevalence, but should explore potential interventions or policies to improve QoWL in healthcare settings.
While acknowledging limitations, the authors could suggest future research directions more explicitly, such as longitudinal designs to establish causality.
Conclusions: The conclusion highlights practical implications but could include a stronger call for policy reforms or interventions based on the study findings.
Technical aspects:
Grammar and syntax: Generally clear, although minor problems with sentence structure and transitions between sections could be polished.
Figures and tables: The tables are well organized, but additional explanatory notes could improve accessibility for readers unfamiliar with the scales.
References: The references are relevant and up-to-date. Including more diverse international studies could provide a broader perspective.
Comments on the Quality of English LanguageVery interesting article, can be published after addressing comments:
This article provides important insights into the intersection of nursing environments and breastfeeding outcomes. Refining the discussion and providing actionable recommendations would further enhance its impact. The study makes a solid contribution to the pediatric and nursing literature, and offers a basis for policy development and clinical practice improvements.
Strengths of the article:
Relevance and importance: The study addresses a critical health issue by examining how the nursing work environment influences breastfeeding prevalence, a significant determinant of maternal and child health.
Comprehensive scope: Inclusion of multiple children's hospitals improves the generalizability of the findings.
Validated instruments: The use of established instruments (e.g., QWI, HSE-IT, and NQoLs) strengthens the reliability of the data.
Analytical rigor: The combination of descriptive statistics, Pearson correlations, and multivariate regression analysis is methodologically sound.
Ethical considerations: The study adheres to the Declaration of Helsinki and ensures anonymity and informed consent, while demonstrating ethical rigor.
Suggestions for improvement:
Abstract: The abstract concisely presents the findings, but it could better outline the limitations and implications for practice to provide a holistic summary.
Introduction: While the introduction is informative, adding more recent global and regional data on breastfeeding trends could better contextualize the study.
Literature review: The background section is strong but could benefit from a more critical appraisal of previous studies that have conflicting findings or gaps in addressing the impact of the nursing work environment.
Methods: The sampling strategy relies on convenience sampling, which may introduce bias. Justification for this approach or discussion of its limitations could strengthen the methodology section.
Results:
The tables are clear but could benefit from additional visual aids (e.g., bar graphs or scatter plots) to illustrate key trends and relationships.
Discussion of differences in breastfeeding prevalence across clinical settings could delve into potential cultural or systemic factors influencing outcomes.
Discussion:
The authors highlight the positive association of QoWL with breastfeeding prevalence, but should explore potential interventions or policies to improve QoWL in healthcare settings.
While acknowledging limitations, the authors could suggest future research directions more explicitly, such as longitudinal designs to establish causality.
Conclusions: The conclusion highlights practical implications but could include a stronger call for policy reforms or interventions based on the study findings.
Technical aspects:
Grammar and syntax: Generally clear, although minor problems with sentence structure and transitions between sections could be polished.
Figures and tables: The tables are well organized, but additional explanatory notes could improve accessibility for readers unfamiliar with the scales.
References: The references are relevant and up-to-date. Including more diverse international studies could provide a broader perspective.
Author Response
Comments 1: General Impression
This very interesting article, can be published after addressing the comments:
This article provides important insights into the intersection of nursing environments and breastfeeding outcomes. Refining the discussion and providing actionable recommendations would further enhance its impact. The study makes a solid contribution to the pediatric and nursing literature and offers a basis for policy development and clinical practice improvements.
Strengths of the article:
Relevance and importance: The study addresses a critical health issue by examining how the nursing work environment influences breastfeeding prevalence, a significant determinant of maternal and child health.
Comprehensive scope: The inclusion of multiple children's hospitals improves the generalizability of the findings.
Validated instruments: The use of established instruments (e.g., QWI, HSE-IT, and NQoLs) strengthens the reliability of the data.
Analytical rigour: The combination of descriptive statistics, Pearson correlations, and multivariate regression analysis is methodologically sound.
Ethical considerations: The study adheres to the Declaration of Helsinki and ensures anonymity and informed consent while demonstrating ethical rigour.
Response 1: Thank you for taking the time to review the article, we really appreciated your suggestions. The suggested changes certainly made a great contribution to the manuscript.
Comments 2: Suggestions for improvement:
Abstract: The abstract concisely presents the findings, but it could better outline the limitations and implications for practice to provide a holistic summary.
Introduction: While the introduction is informative, adding more recent global and regional data on breastfeeding trends could better contextualize the study.
Literature review: The background section is strong but could benefit from a more critical appraisal of previous studies that have conflicting findings or gaps in addressing the impact of the nursing work environment.
Methods: The sampling strategy relies on convenience sampling, which may introduce bias. Justification for this approach or discussion of its limitations could strengthen the methodology section.
Results: The tables are clear but could benefit from additional visual aids (e.g., bar graphs or scatter plots) to illustrate key trends and relationships.
Discussion of differences in breastfeeding prevalence across clinical settings could delve into potential cultural or systemic factors influencing outcomes.
Discussion: The authors highlight the positive association of QoWL with breastfeeding prevalence, but should explore potential interventions or policies to improve QoWL in healthcare settings.
While acknowledging limitations, the authors could suggest future research directions more explicitly, such as longitudinal designs to establish causality.
Conclusions: The conclusion highlights practical implications but could include a stronger call for policy reforms or interventions based on the study findings.
Response 2: Thank you for your suggestions for improvement. We provide the changes you indicated:
- Abstract: we have now specified that data collection was limited to low and medium-intensity care. Our implications are: “To enhance breastfeeding support during child hospitalizations, healthcare policies should address workload management, stress reduction, and work-life balance.”
We have also added a limitation to address in the future: “Future research should expand to other clinical settings and include detailed patient age data to understand this relationship”.
- Introduction: We have now revised the introduction to be more specific.
- Literature review: We have now revised this section to emphasize the existing gap in the literature.
- Methods: We have now addressed the limit of convenience sampling in the limitations. Also, we have added a sample size estimation.
- Discussion: The discussion has been revised according to your suggestion.
- Conclusion: The conclusion has been revised and a stronger call for policy reforms has been written.
Comments 3: Technical aspects:
Grammar and syntax: Generally clear, although minor problems with sentence structure and transitions between sections could be polished.
Figures and tables: The tables are well organized, but additional explanatory notes could improve accessibility for readers unfamiliar with the scales.
References: The references are relevant and up-to-date. Including more diverse international studies could provide a broader perspective.
Response 3: Thanks to your comment, a language revision was carried out with a native speaker expert and changes were made in the text to the tables to make them more explanatory
Comments 4: Comments on the Quality of English Language
This very interesting article, can be published after addressing the comments:
This article provides important insights into the intersection of nursing environments and breastfeeding outcomes. Refining the discussion and providing actionable recommendations would further enhance its impact. The study makes a solid contribution to the pediatric and nursing literature and offers a basis for policy development and clinical practice improvements.
Strengths of the article:
Relevance and importance: The study addresses a critical health issue by examining how the nursing work environment influences breastfeeding prevalence, a significant determinant of maternal and child health.
Response 4: Thank you for your valuable comments. We hope that the manuscript is now clear.
Round 2
Reviewer 2 Report
Comments and Suggestions for Authors
The author(s) have made all suggested changes and suggestions for improvement. The work is of acceptable quality.